# Highly Selective Adsorption on SiSe Monolayer and Effect of Strain Engineering: A DFT Study

**DOI:** 10.3390/s20040977

**Published:** 2020-02-12

**Authors:** Quan Zhou, Lian Liu, Qipeng Liu, Zeping Wang, Chenshan Gao, Yufei Liu, Huaiyu Ye

**Affiliations:** 1Key Laboratory of Optoelectronic Technology & Systems, Education Ministry of China, College of Optoelectronic Engineering, Chongqing University, Chongqing 400044, China; LucyZhou@cqu.edu.cn (Q.Z.); lianliu@cqu.edu.cn (L.L.); cowboy.lqp@cqu.edu.cn (Q.L.); 20170801032@cqu.edu.cn (Z.W.); gao_chenshan@cqu.edu.cn (C.G.); yufei.liu@cqu.edu.cn (Y.L.); 2Centre for Intelligent Sensing Technology, College of Optoelectronic Engineering, Chongqing University, Chongqing 400044, China; 3School of Microelectronics, Southern University of Science and Technology, Shenzhen 518055, China; 4Shenzhen Institute of Wide-Bandgap Semiconductors, No.1088, Xueyuan Rd., Xili, Nanshan District, Shenzhen 518055, Guangdong, China

**Keywords:** SiSe monolayer, sensor, DFT, adsorption, stress

## Abstract

The adsorption types of ten kinds of gas molecules (O_2_, NH_3_, SO_2_, CH_4_, NO, H_2_S, H_2_, CO, CO_2_, and NO_2_) on the surface of SiSe monolayer are analyzed by the density-functional theory (DFT) calculation based on adsorption energy, charge density difference (CDD), electron localization function (ELF), and band structure. It shows high selective adsorption on SiSe monolayer that some gas molecules like SO_2_, NO, and NO_2_ are chemically adsorbed, while the NH_3_ molecule is physically adsorbed, the rest of the molecules are weakly adsorbed. Moreover, stress is applied to the SiSe monolayer to improve the adsorption strength of NH_3_. It has a tendency of increment with the increase of compressive stress. The strongest physical adsorption energy (−0.426 eV) is obtained when 2% compressive stress is added to the substrate in zigzag direction. The simple desorption is realized by decreasing the stress. Furthermore, based on the similar adsorption energy between SO_2_ and NH_3_ molecules, the co-adsorption of these two gases are studied. The results show that SO_2_ will promote the detection of NH_3_ in the case of SO_2_-NH_3_/SiSe configuration. Therefore, SiSe monolayer is a good candidate for NH_3_ sensing with strain engineering.

## 1. Introduction

As people care more about physical health and living environment, gas dectection [1,2], especially for the harmful gases has attracted much attention [3,4,5,6,7], such as the monitoring of indoor formaldehyde [8,9] gas content or the leakage of ammonia [10] in chemical plants. The detector can warn in advance when the gas leaks or is excessive. Thus, the gas sensor [11,12] requires sensitive materials which have ability to convert gas information into electrical signals, such as potential or resistance [13].

The principle of gas sensor includes chemical adsorption and physical adsorption. Chemisorption has strong interaction and a slow reaction rate caused by the chemical bond between adsorbent and adsorbate. On the contrary, physical adsorption is weak and faster caused by intermolecular Van der Waals attraction without chemical reaction and is easy to recover. Physisorption and chemisorption are not absolutely isolated from each other and can be changed from one to another. Adsorption properties of sensing materials can be adjusted through a variety of physical or chemical methods [14]. The weak interaction between adsorbate and adsorbent can be enhanced by heating [4], doping [15,16], and even possibly transformed into chemisorption, while strong adsorption can be weakened under the impact of charge injection [10] and electric field [17]. In addition, original chemical adsorption is converted to physical adsorption [18], which is in favor of gas sensing. However, the above-mentioned modification methods are characterized as instable and complex. Recently, Yu et al. have calculated that chemical adsorption of NH_3_ by Ti_2_CO_2_ substrate is improved by stress [19]. Ma et al. have explained that the moderate adsorption of SO_2_ was further enhanced by applying tensile stress [20]. Park et al. prepared the MoS_2_ bilayer on substrate by chemical vapor deposition (CVD) and utilized the load test machine of strain gauge presenting different strength [21]. Therefore, stress is a simple, effective, and suitable method for the adjustment of gas adsorption.

For high sensitivity detection, novel two-dimensional (2D) materials are considered to have great potential due to the quantum size effect [22] and exposed active sites. For example, graphene [23,24,25] or single-layer phosphorene [26,27,28] has attracted much attention because of high selectivity and sensitivity in adsorption behavior. In particular, the black phosphorus gas sensor has been confirmed by experiments to be a good detector for NO_2_ gas molecule [29]. The pleated structure of phosphorene is susceptible to strain [30,31] and its electrical properties can be easily regulated through stress. However, inherent defects of phosphorene and instability in air hinder its further development. Recently, the SiSe monolayer [32,33], with similar structure of phosphorene has shown higher stability than phosphorene. Sa et al. proposed that the good stability of SiSe material is an ideal candidate of building block structure [34]. Yang et al. demonstrated that SiSe monolayers made from non-toxic and earth-abundant elements intrinsically have low thermal conductivities [35]. It is also reported that SiSe can be applied in a battery [36,37] due to large adsorption energies towards single Na and Li atoms, on account of the lower activation energy and rich active sites. These properties may also be expected to be utilized in gas sensor. Especially, an SiSe monolayer with abundant sources in the earth possesses a tunable electronic structure [38] and flexibility under stress, which is conducive to gas adsorption.

Herein, based on the density-functional theory (DFT), adsorption mechanisms and adsorption types of SiSe monolayer with different molecules (O_2_, NH_3_, SO_2_, CH_4_, NO, H_2_S, H_2_, CO, CO_2_, and NO_2_) are studied. From the perspective of structure and electronic properties, the behavior of NH_3_ molecules on the SiSe substrate is physical adsorption with moderate adsorption energy, while SO_2_, NO, and NO_2_ molecules on SiSe substrate show chemical adsorption with larger adsorption energy. Further, stress is used to adjust the adsorption performance. Results show that the adsorption of NH_3_ works best when 2% compressive stress is added to the substrate in zigzag direction, maintaining the maximum physical adsorption characteristics. It is also proved that the NH_3_ gas can be desorbed when the stress is removed. By contrast, the SO_2_ system exhibits stronger chemical adsorption under compressive stress. Then co-adsorption of SO_2_ and NH_3_ gas has also been studied due to the interference of each other. We find that the SO_2_ molecule promotes the adsorption of NH_3_ to some extent, which is advantageous as an NH_3_ gas sensor.

## 2. Materials and Methods

In this work, the DFT theory calculations is conducted by using the CASTEP module and Dmol^3^ package of Materials Studio, with generalized gradient approximations (GGA) [39] to Perdew-Burke-Ernzerhof (PBE) [40] functional for the exchange-correlation term [41]. In addition, the Grimme method is used as a dispersion correction to describe the interlayer Van der Waals interaction [42]. The systems are constructed with 3 × 3 × 1 SiSe supercell as substrate and gas molecules are placed above the monolayer. Due to the difference between polar and non-polar molecules, non-spin polarization method is used in (CO_2_, NH_3_, CO, H_2_S, CH_4_, SO_2_, and H_2_O)/SiSe adsorption systems, and spin polarization calculation is used in (NO_2_, O_2_, and NO)/SiSe adsorption systems. The vacuum layer is set to 20 Å in the (0 0 1) lattice plane to reduce periodic interaction between the layers in the vertical direction. In our simulation, the energy cutoff of the plane wave base is set as 500 eV. The brillouin zone integration is used for geometric optimization and electronic performance calculation on 16 × 16 × 1 grid. All atomic coordinates and lattice constants gradually relax until a force converges within 0.01 eV Å^−1 ^per atom and the tolerance of energy is less than 1.0 e^-5^ eV per atom. The adsorption energy is defined in the following equation [43]:
*E_a_* = *E_SiSe+gas_* − *E_SiSe_* − *E_gas_*,(1)
where *E_SiSe+gas_*, *E_SiSe_,* and *E_gas_* represent the total energy adsorbed by the gas on the SiSe substrate, the SiSe monolayer, and the gas molecules, respectively. Here we mainly focus on absolute value, the greater the absolute value of the adsorption energy, the stronger the interaction between the SiSe monolayer and the gas molecules.

The charge density difference (CDD) map is described to see the charge distribution of gas molecules with the substrate visually and qualitatively. The CDD is based on following definition with charge analysis [44]:
*ρ* = *ρ_gas +SiSe_* − *ρ_SiSe_* − *ρ_gas_*,(2)

In this formula, *ρ_SiSe+gas_*, *ρ_SiSe_*, and *ρ_gas_* represent the total charge density of the gas molecule/SiSe system, the original SiSe monolayer, and the isolated gas molecules. During the analysis, we need to be aware that each component should be in the same position in the adsorption structure.

Electron localization function (ELF) represents the electron location function and depicts the localized distribution characteristics of electrons. By definition, the value of ELF ranges from 0 to 1, marked with blue and red in the map, respectively. When the value is close to 1, this means the electrons of the atoms are highly localized and bonded together.

## 3. Results and Discussion

### 3.1. Adsorption on SiSe Monolayer

It begins with the geometry optimization of unit cell of SiSe. Then 3 × 3 × 1 surface cell is constructed, including 9 Si atoms and 9 Se atoms in the surface cell. After full relaxation, the lattice constant of the armchair direction is 3.650 Å (marked “a” in Figure 1), and the lattice constant of the zigzag direction is 4.647 Å (marked “b” in Figure 1) [38], which is similar to the structure of black phosphorus (4.57 Å and 3.51 Å respectively) [45]. The covalent bond length between Si-Se is tested to be 2.495 Å in horizontal direction (marked “l_H(Se-Si)_” in Figure 1) and 2.467 Å in vertical direction (marked “l_V(Se-Si)_” in Figure 1), which is consistent with previous values [36]. Then, in analyzing the optimal adsorption position displayed in Figure 1, four different adsorption positions are considered, including the “A” position (right above Si-Se bond), the “B” position (on the top of the Si atom), “C” position (right above the center of the honeycomb area), and the “D” position (on the top of the Se atom). The different molecular orientations are also considered in Appendix A. All calculations and analyses are based on systems that the gas molecules at 3 Å away from four sites [43]. After the four configurations converged, we select one configuration with the largest absolute value of the adsorption energy as the object of the next calculation, which is listed in Appendix A. Similarly, other adsorption systems are calculated in the same way. The optimized site of each adsorption system is shown in Table 1.

To discuss the difference in adsorption of different gases on SiSe monolayer, the adsorption energy and distance are calculated (seeTable 1). After full relaxation, NO and NO_2_ configurations manifest the biggest adsorption energy (|−0.734| eV and |−0.735| eV, respectively) with the shortest adsorption distance (1.835 Å and 1.995 Å) among all the systems. In addition, the Se-Si bond enlarges to 2.608 Å in the vertical direction and 2.409 Å in the horizon direction, which are attributed to the strong interaction with the substrate. By contrast, compared with other gas configurations, CO, O_2_, CO_2_, and H_2_S molecules have repulsive interaction with the substrate, judged from their increased adsorption distance than the initial state. Other gas molecules like H_2_O are preliminarily considered as weak interaction with the substrate based on low adsorption energy. Specially, NH_3_ and SO_2_ configurations have moderate adsorption energy values (|−0.414| eV and |−0.489| eV, respectively). The distance between adsorbates and adsorbents is reduced to 2.471 Å and 2.733 Å, respectively. The Si-Se bond is slightly distorted, whose adsorption mechanisms are obscure and need further analysis.

In order to clarify the interaction between molecules and SiSe monolayer, the CDD map and ΔQ are calculated to see the charge distribution of gas molecules with the substrate qualitatively and quantitatively. We have listed some typical CDD maps that reflect the different degrees of interaction between the substrate and the gas molecules. The incremental tendency of adsorption strength is displayed from Figure 2a to Figure 2f. In Figure 2a, the CO configuration has substantially no accumulation and consumption of charge at the interface with a little ΔQ, and the same situation happens in the CO_2_ configuration (see Appendix A). In other systems, such as O_2_, CH_4_ etc., the substrate and the gas have a weaker charge rearrangement as shown in Figure 2b, which manifests the stronger interaction with the substrate than CO configuration. Yet the interaction is still much smaller. In Figure 2c,d, the range of charge redistribution of NH_3_ configuration and SO_2_ configuration are distinctly expanded. The charge is more severely depleted on the NH_3_ molecule, while the accumulation on the SO_2_ molecule is more serious, indicating that there is a strong interaction between the gas molecules and the substrate. Correspondingly, ΔQ of these two systems is an order of magnitude larger than other six weak systems mentioned above. In the Figure 2e,f, the interaction between NO and NO_2_ structures still maintain the strongest position—the maximum range of charge rearrangement and a significant charge accumulation consumption. It shows, apparently, the loss of charge locates on the substrate, while the increase in charge is concentrated on the gas molecules, where charge is transferred from the gas molecule to the SiSe monolayer. In addition, the stronger interaction matches bigger charge transfer value (|-0.258| e) in NO_2_ configuration. It is worth noting that charge transfer of SO_2_ molecule shows an abnormal change (|-0.238| e), increasing sharply compared to the NH_3_ molecule, even bigger than the NO molecule (|-0.220| e), which implies a different adsorption type from NH_3_ configuration.

Associated with ELF map in Figure 3a, it is clear to notice that the value of ELF between CO and SiSe substrate is close to 0. There are hardly any electrons between CO molecule and SiSe substrate, which means no electron sharing and no covalent bonds exist in these two components. The same situation also occurs in systems like CH_4_, H_2_O, H_2_S, and O_2_ (see Appendix A), which shows that these gas configurations are all physically adsorbed. In Figure 3b, the value of ELF in the border of NH_3_ and SiSe monolayer is nearly 0.5, revealing that most electrons are still highly concentrated on the gas molecules and substrate, very little are delocalized in edge of junction. That is to say there exist very few electrons between NH_3_ and SiSe monolayer and it is not sufficient to form chemical bonds. On the contrary, it is distinct to see that electron densities of S and Si atoms are connected together and electrons are highly located between them in Figure 3c. It is highly possible to form S-Si covalent bond in the system. In the case of NO_2_ system in Figure 3d, the serious orbital overlaps between N atoms and Si atoms displays that they are bonded together, which reflects the chemical adsorption of NO and NO_2_ again.

To further confirm the adsorption type, the DOS and PDOS (total and partial densities of states) are calculated.Figure 4a shows plainly that DOS of pristine SiSe substrate is almost changeless near Fermi level before and after the NH_3_ is adsorbed on it. The big difference appears at around −5 eV in the valence band far away from Fermi level. In PDOS, it is obvious to find that there is no orbital overlap between NH_3_ molecule and SiSe monolayer near CBM (Conduction Band Minimum) or VBM (Valence Band Maximum). Hence physical adsorption occurs in NH_3_ and SiSe monolayer. In next Figure 4b, the DOS of SiSe is slightly offset after SO_2_ is adsorbed on SiSe near Fermi level from -2 eV to -3 eV, and the peaks of orbitals between S atom and Si atom are overlapped at the energy of -1 eV in the valence band and 1 eV in the conduction band. From Figure 4c,d, between −8 eV to 4 eV, the DOS of pristine SiSe monolayer is different from the adsorbed systems. The original electrical properties are destroyed when the NO_2_ is adsorbed on substrate. The chemical bond is formed with the more serious orbitals’ hybridization, combined with the above analysis.

In total, the high selectivity is displayed based on above analysis associated with the electronic analysis: the interaction between the NH_3_ gases and the SiSe monolayer is the strongest physical adsorption property with moderate adsorption energy (−0.414 eV), which is larger than previous researches [4,19,27], and makes it possible for SiSe monolayer to adsorb NH_3_ and easily desorb. SO_2_, NO or NO_2_ gases on the SiSe monolayer show the chemisorption property. The chemical reaction with substrate makes it difficult to reuse. The rest of the gases have very weak interaction with SiSe monolayer. They are all not suitable as a gas sensor based on the SiSe monolayer.

### 3.2. Adsorption Mechanism of NH_3_ Under Stress

Due to the special corrugated structure of SiSe, the adsorption property with SiSe monolayer under stress is discussed. Stress has been used to effectively regulate the mechanical, electrical, and magnetic properties of semiconductors over the past researches [46,47,48,49]. Previous theoretical and experimental studies indicate that the application of in-plane strain can adjust the band gap of SiSe monolayer [38], yet the performance of the SiSe monolayer adsorbing gas after the stress has not been studied. NH_3_ needs to be detected as a harmful gas, stress calculations are performed along the X-axis (zigzag direction), Y-axis (armchair direction), and biaxial directions, respectively (shown in Figure 5). The stress is defined as: ε = (a − a_0_)/a_0_, where “a_0_” and “a” are the lattice parameter of the original cell and strained-cell, respectively [19]. The gas is adsorbed on systems with the substrate stretched or compressed from 0% to 8% (also recorded −8% to 8%), the changes in various indexes after adsorption are observed.

#### 3.2.1. Compressive Direction

First, under the uniaxial X-axis compression, the adsorption energy is gradually increased. When the stress reaches −8%, the adsorption energy increases to a maximum value of |−0.46| eV, which indirectly reflects the enhancement of the interaction between the NH_3_ molecule and the substrate. Further, this enhancement may result in the formation of covalent bonds between the NH_3_ molecule and the substrate. Therefore, the research on the ELF chart is performed. As present in Figure 6a, the electron local density of the NH_3_ molecules does not overlap with the SiSe substrate in ELF slice until the compressive stress reaches to 4%, and it is seen that a covalent bond is likely to be formed, which is in the progressing direction that is unfavorable to the gas sensor. Thus, it shows the best adsorption efficiency when −2% stress is added to the substrate. Similarly, the compressive strain trend in the Y-axis and the biaxial are the same as the compressive strain trend in the X-axis (as shown in Figure 6b). Only the degree of change is different: biaxial > Y-axis > X-axis. When the stress is from −4% to −8%, the rate of change in adsorption energy is sharply increasing and the adsorption energy increases to |−0.78| eV at −8%, which is greater than the adsorption energy of NO_2_ system. At this time, observing the ELF diagram, it can be seen that the electron local density range of the N and Si atom are greatly hybridized (see inset of Figure 6b). A strong chemical bond is formed between the N and the Si atoms. That is to say, chemical adsorption is formed between the substrate and the gas, and the compressive stress enhances the adsorption of the gas.

According to the reversibility of physical adsorption, we try to demonstrate the inverse process of the best adsorption position when −2% pressure is applied to SiSe monolayer. The −2% compression system that has completed the adsorption of NH_3_ is pulled back again to the original state of 0% stress-free state. After optimization, the adsorption energy, closest distance, ΔQ and the bond of Si-Se are recorded in Table 2. Comparing the pulled back to the 0% system (label as −2% to 0% NH_3_/SiSe) with −2% compressive stress-NH_3_/SiSe (label as −2% NH_3_/SiSe), it is discovered that adsorption energy is reduced to |−0.413| eV, which is very close to |−0.414| eV at 0%. Additionally, the length of the Si-Se bond and ΔQ both recover to the situation of no stress, which shows an reverse process of gas desorption in gas sensor.

In the process of applying compressive stress to the substrate, the NH_3_ molecule is found to move close to the Si atom in Figure 6c,d. Moreover, the bond length of Si-Se, ΔQ, and adsorption distance of the adsorption process of biaxial stress are carefully observed in Table 3 due to the obvious change among them. Without compression, after the adsorption is stabilized, the length of Si-Se bond are slightly distorted to 2.502 Å in XY plane and 2.658 Å in YZ plane respectively, the NH_3_ molecule will run toward the middle of the corrugate with the Si atom attracted to protrude from the substrate level, causing the substrate to be slightly deformed. As the biaxial compression increases, the l_H(Se-Si)_ gradually decreases to 2.436 Å and 2.444 Å. While the l_V(Se-Si)_ gradually increases to 2.701 Å, meanwhile the adsorption distance is gradually reduced to 2.226 Å, representing the strengthen interaction between the substrate and NH_3_ molecule. At this time, the NH_3_ molecules no longer run to the middle of the corrugate, but is more close to the Si atom without the silicon atoms prominent out of the horizon. This phenomenon can be explained by charge transfer. When there is no pressure, the ΔQ of Si atom (|0.605| e) and Se atom (|−0.618| e) close to the NH_3_ molecule is very large. It can be seen that the NH_3_ molecule is not only attracted by the Si atom, but also restricted by the Se atom, so that the NH_3_ molecule will run to the middle, making Si-Se length in the vertical direction increases. As the pressure is applied, the ΔQ of the Si atom (|0.672| e) is gradually larger than that of the Se atom (|−0.563| e), so that the attraction of the Si atom is dominant. Thus, the NH_3_ molecule is adsorbed towards the Si atom.

#### 3.2.2. Tensile Direction

For NH_3_ configuration, the adsorption energy of the system decreases very little as the tensile force increases, which is contrary to the compression. The adsorption energy in the direction of X-axis slightly decreases to |-0.396| e with the stretch increases, and the Y-axis direction decreases to the minimum value of |-0.313| e with the stretch increases to 4%, then conversely increases to |-0.405| e at 8%. It is noticed that the tensile stress in the Y-axis direction is more significant to the system, and the biaxial is the effect of the X-axis and the Y-axis. As the adsorption energy decreases, the effect of gas and substrate becomes weaker. Combined with ELF in Figure 7a, the range of NH_3_ molecule under biaxial stretching does not overlap with the SiSe monolayer, indicating no covalent bond is formed, thus the physical adsorption property remains.

When the tensile stress is added to the substrate, the NH_3_ molecule moves to the honeycomb instead in Figure 7c,d. The mechanism is studied from the perspective of charge transfer and atomic orbital. First, it can be seen from the ΔQ of the substrate that the Si atom closest to the NH_3_ molecule has a large ΔQ and there is a strong interaction with the NH_3_ molecule mainly. However, the two Se atoms connected to the Si atom also have large ΔQ (|−0.618| e) that cannot be ignored. In total, the entire NH_3_ molecule is simultaneously constrained by these relatively active Se atoms. As the stretching enlarges, the ΔQ of Se atom increases to |−0.627| e at 8% with the enhancement of constraint action, so that the entire NH_3_ molecule is attracted to the middle of the honeycomb. Hence, the Si atom follows the movement of the NH_3_ molecule and the Se-Si bond in the vertical direction becomes longer inTable 4.

In the following, the band structure and atomic orbital are discussed to have an in-depth exploration. In the absence of stress displayed in Figure 8a, the system has an indirect band gap of 1.108 eV. Presented in PDOS, the valence band is mainly contributed by the 4p orbital of the Se atom, and the conduction band is mainly contributed by the 3p orbital of the Si atom, which is in accordance with the HUMO (Highest Occupied Molecular Orbital) and LUMO (Lowest Unoccupied Molecular Orbital) in Figure 8c,d. As the tensile force increases, the bandgap of the system increases to 1.295 eV. This may be the reason that the tensile force increases the lattice constant of the substrate and the distance between the atoms becomes larger, thus the energy level of each atom becomes more discrete. The contribution of the P orbital of Se to the conduction band increases, and the effect of Se increases under the tensile force, which is consistent with the previous analysis. At the same time, the contribution of the Si atom to the valence band has also increased, but there is not much increase in Se, and finally the comprehensive influence of the substrate on the adsorption of NH_3_ is weakened.

### 3.3. Adsorption Mechanism of SO_2_ Under Stress

Since the adsorption of SO_2_ molecule by the SiSe monolayer is very close to the adsorption of NH_3_, it is hoped that the adsorption of SO_2_ molecule can be improved by stress, too. Similarly, stress is applied to the substrate from the X-axis, the Y-axis, and the XY-biaxial direction, respectively, and then the SO_2_ molecule is adsorbed. The trend of adsorption strength is also the same as NH_3_ configuration: the increase of the tensile force causes the decrease of adsorption energy and the ΔQ, while the increase of the pressure causes the increase of adsorption energy and the ΔQ. It has been previously analyzed that the adsorption of SO_2_ by the substrate with free-stress is chemisorption, and if the stretching method is employed to reduce the interaction between the substrate and the gas, it is possible to improve the adsorption of SO_2_ molecule for gas sensor. Furthermore, the ELF of SO_2_ with 8% stress added to the SiSe monolayer is explored. When the tensile stress of X-axis is increasing, the adsorption energy reduces by |0.073| eV and the ΔQ reduces by 0.32 e, showing a declining interaction between the substrate and molecule. However, the region of electron localization between the Si atom and the S atom still overlaps largely in Figure 9a, proving that chemisorption is still present. Although the adsorption energy and the transfer charge reduce more under the tensile stress of Y-axis or biaxial, such as the adsorption energy reaches |−0.400| eV and the ΔQ reaches |−0.208| e under biaxial stretching, but the ELF slice still demonstrates the hybridization between two parts. It is to say that the stress cannot adjust the adsorption of SO_2_ to an optimum state.

On account of the small difference in adsorption of NH_3_ and SO_2_ based on SiSe substrate. It is necessary to discuss the co-adsorption mechanism of SO_2_ and NH_3_ gas on SiSe monolayer [50]. We consider three possible adsorption manners according to the actual situation: adsorption of SO_2_ and NH_3_ simultaneously (referred to as SO_2_ and NH_3_/SiSe), adsorption of SO_2_ followed by adsorption of NH_3_ (referred as SO_2_-NH_3_/SiSe) and adsorption of NH_3_ followed by adsorption of SO_2_ (referred as NH_3_-SO_2_/SiSe). The results of the calculation are shown in the Table 5 and Appendix A. Under these three cases, the substrate exhibits a different strength for gas adsorption. In the case of SO_2_ and NH_3_/SiSe, the adsorption abilities to the SO_2_ molecule and the NH_3_ molecule both increase from the data analysis in the table. The ΔQ of NH_3_ rises to |0.241| e, which is |0.063| e larger than initial. Meanwhile the ΔQ of SO_2_ is 0.046 e larger than original. Both the distances between the molecules and substrate are shortened, with the NH_3_ reduced more seriously. It is easy to find that the same situation occurs in NH_3_-SO_2_/SiSe. Hence, the co-adsorption of NH_3_ in the case of SO_2_ and NH_3_/SiSe and NH_3_-SO_2_/SiSe shows no sign of decline. Differently, when the SiSe monolayer first adsorbs the SO_2_ molecule and then adsorbs the NH_3_ molecule, the ΔQ of SO_2_ decreases with the adsorption distance of SO_2_ increases comparing to the SO_2_/SiSe system, indicating that the interaction between the SO_2_ molecule and substrate is weakened. The adsorption of NH_3_ gas is improved comparing to NH_3_/SiSe system, which may due to the reason that electronegativity of the N atom is greater than S atom. The N and Si atom are more likely to get or lose electrons than the S and Si atom.

## 4. Conclusions

In conclusion, ten different gases (CO, O_2_, CO_2_, CH_4_, H_2_S, H_2_O, NH_3_, SO_2_, NO, NO_2_) of adsorption on SiSe monolayer have been studied. From the perspective of adsorption energy, DOS, PDOS, CCD, and ELF, it is found that NH_3_ molecule is physical adsorption with moderate adsorption energy of 0.414 eV, which is the most suitable as gas sensor material. On the contrary, the adsorption of SO_2_ belongs to chemical adsorption with large adsorption energy (0.490 eV), both NO and NO_2_ are equipped with the largest adsorption energy (0.734 eV, 0.735 eV), which can be applied in one-time gas sensor. Further, stress is applied to adjust the adsorption performance of SiSe monolayer to adsorb NH_3_ and SO_2_ molecules. The results reveal that compressive strain causes the sensitive material sensor to develop toward larger adsorption energy, and the tensile strain changes inversely. Thus, when the stress increases to −2% in the X-axis, the adsorption of NH_3_ is enhanced with the adsorption energy increases to |−0.426| eV, which maintains the physical adsorption of NH_3_. While for SO_2_ molecule, the strength of chemical adsorption based on SiSe monolayer is increased under compressive stress. In addition, due to the small difference of SO_2_ and NH_3_ molecules, the co-adsorption of these two gases on substrate shows that the adsorption of NH_3_ is promoted at three different situations, even the adsorption of SO_2_ will be suppressed when the SO_2_ is first adsorbed on SiSe monolayer. Therefore, it is expected that the SiSe monolayer is a potential candidate to be applied in NH_3_ gas sensors.

## Figures and Tables

**Figure 1 sensors-20-00977-f001:**
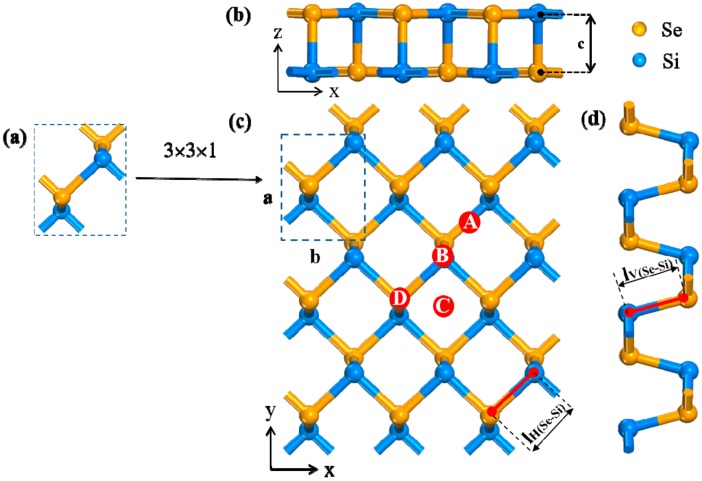
(**a**) Unit cell, (**b**) front, (**c**) top, and (**d**) side view of (0 0 1) lattice plane of substrate. The four sites “**A**”, “**B**”, “**C**”, and “**D**” are respectively labeled in the top view. “x”, “y”, and “z” are the lattice directions. The marked “**a**” and “**b**” represent the lattice constant of unit cell.

**Figure 2 sensors-20-00977-f002:**
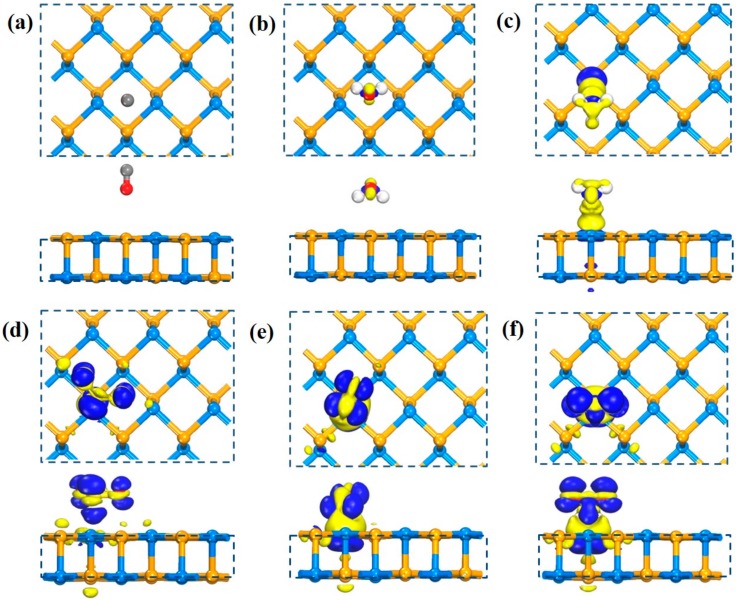
Front and top charge distribution view of charge density difference (CDD) maps, in turn are (**a**) CO, (**b**) H_2_O, (**c**) NH_3_, (**d**) SO_2_, (**e**) NO, (**f**) NO_2_ configurations. The yellow part indicates charge loss, and the blue part indicates charge accumulation. The isosurface is set as 0.01e/Å^3^.

**Figure 3 sensors-20-00977-f003:**
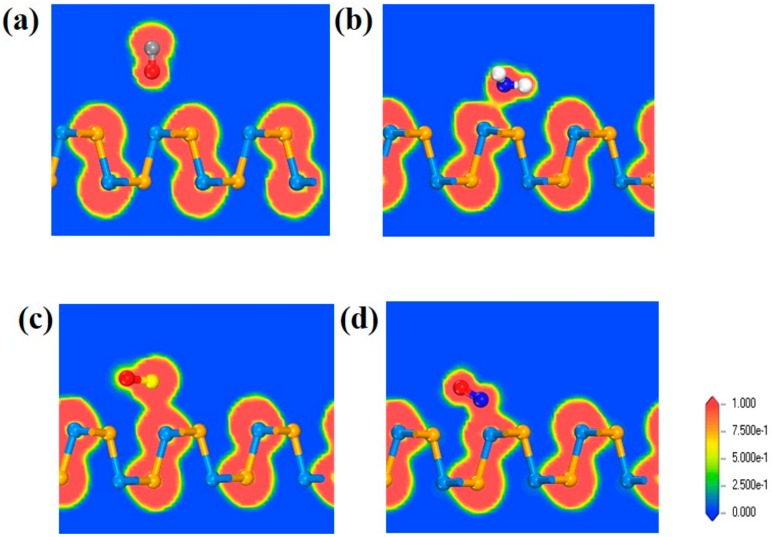
The distribution of electron localization maps of (**a**) CO, (**b**) NH_3_, (**c**) SO_2_, and (**d**) NO_2_ configurations. The reference column for the electron localization function (ELF) value from 0 to 1 is located on the right side of the figure, representing blue and red, respectively. The slice of the ELF is parallel to the (100) crystal plane.

**Figure 4 sensors-20-00977-f004:**
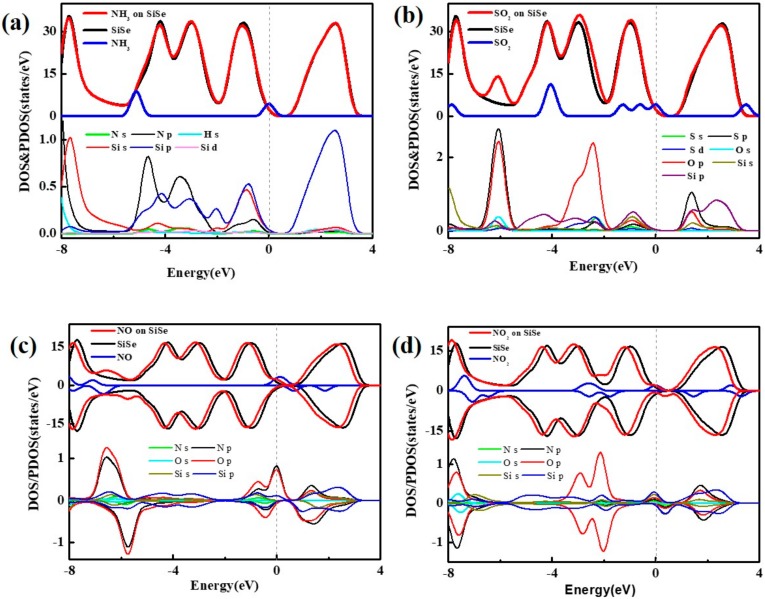
The DOS and PDOS maps of (**a**) NH_3_, (**b**) SO_2_, (**c**) NO, and (**d**) NO_2_ molecules adsorbed on SiSe monolayer.

**Figure 5 sensors-20-00977-f005:**
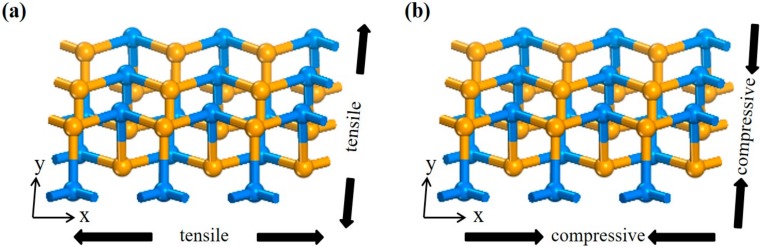
Schematic diagram of SiSe monolayer under in-plane (**a**) compressive (**b**) tensile stress from X-axis, Y-axis, and biaxial direction.

**Figure 6 sensors-20-00977-f006:**
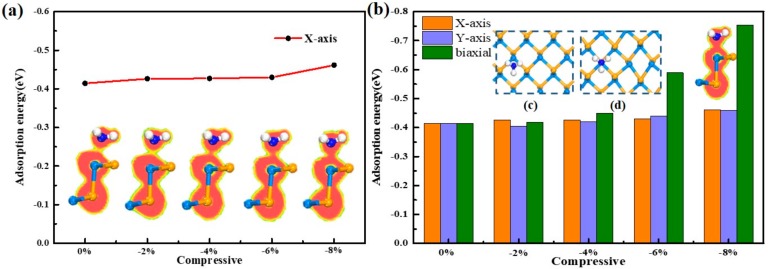
(**a**) The change curves of E*_ad_* with compressive strain from 0% to −8% in NH_3_/strained-SiSe systems, ELF from left to right represents 0%, −2%, −4%, −6%, and −8% of stress. (**b**) The comparison of adsorption energy with X-axis, Y-axis, and biaxial under compressive strain from 0% to −8% in NH_3_/strained-SiSe systems, the data is shown in Appendix A. The inset of (**b**) represents the ELF of −8% biaxial of NH_3_/SiSe monolayer. The inset is morphology of NH_3_ system (**c**) without stress and (**d**) with −8% stress in biaxial direction.

**Figure 7 sensors-20-00977-f007:**
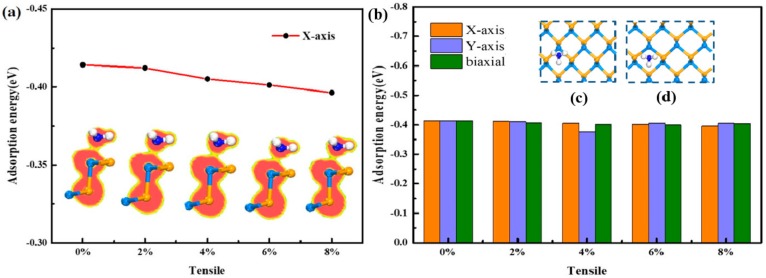
(**a**) The change curves of E*_ad_* with tensile strain from 0% to 8% in NH_3_/strained-SiSe systems, ELF from left to right represents 0%, 2%, 4%, 6%, and 8% of stress. (**b**) The comparison of adsorption energy with the X-axis, Y-axis, and biaxial under compressive strain from 0% to 8% in NH_3_/strained-SiSe systems, the data is shown in Appendix A. The inset is morphology of NH_3_ system (**c**) without stress and (**d**) with 8% stress in biaxial direction.

**Figure 8 sensors-20-00977-f008:**
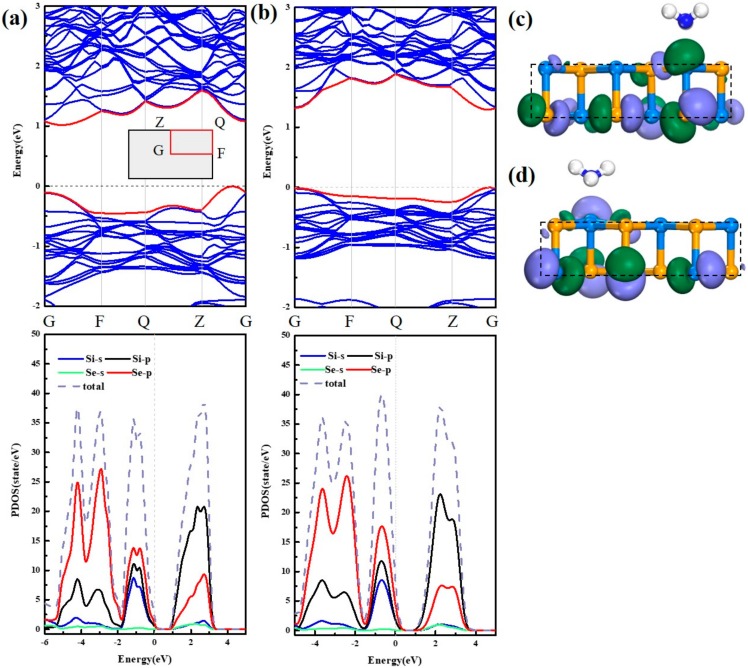
The band structure and PDOS of (**a**) NH_3_/ SiSe systems and (**b**) 8% tensile stress-NH_3_/SiSe systems, respectively. The inset of (**a**) is the Brillouin Zone path. The Fermi level (the dash line) is set to zero. (**c**) HUMO and (**d**) LUMO.

**Figure 9 sensors-20-00977-f009:**
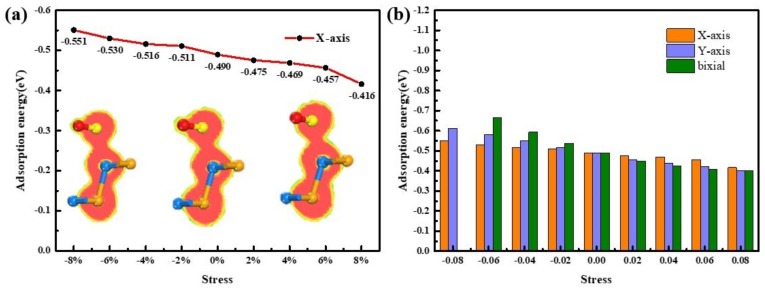
(**a**) The change curves of E*_ad_* with tensile strain from −8% to 8% in SO_2_/strained-SiSe systems, ELF from left to right represents −8%, 0%, and 8% of stress. (**b**) The comparison of adsorption energy with the X-axis, Y-axis, and biaxial under compressive strain from -8% to 8% in NH_3_/strained-SiSe systems.

**Table 1 sensors-20-00977-t001:** The optimal adsorption position, adsorption energy (E_ad_), closest distance (d), charge transfer (ΔQ), and the length of Si-Se bond. The distance is defined as the nearest two atomic center distances between the gas molecules and the SiSe monolayer. ΔQ is from Hirshfled/Mulliken population analysis, a negative value of ΔQ represents that electrons are transferred from the SiSe monolayer to the gas molecule. The two values of SiSe bond represent the horizontal direction (l_H(Se-Si)_)/vertical direction (l_V(Se-Si)_).

Configuration	Site	E_ad_(eV)	d (Å)	ΔQ (e)	Si-Se(Å)
CO	C	−0.071	3.577	0.002/0.008	2.497/2.462
O_2_	B	−0.098	3.067	−0.032/−0.037	2.492/2.467
CO_2_	D	−0.132	3.424	−0.016/-0.016	2.495/2.469
CH_4_	A	−0.133	3.084	−0.066/0.022	2.497/2.463
H_2_S	A	−0.203	3.405	0.068/0.080	2.503/2.497
H_2_O	C	−0.237	2.805	−0.097/0.032	2.487/2.463
NH_3_	B	−0.414	2.471	0.153/0.178	2.503/2.658
SO_2_	A	−0.489	2.686	−0.238/−0.197	2.448/2.441
NO	B	−0.734	1.835	−0.220/−0.355	2.417/2.528
NO_2_	B	−0.735	1.995	−0.285/−0.386	2.409/2.608

**Table 2 sensors-20-00977-t002:** The adsorption energy (E_ad_), closest distance (d), Mulliken charge transfer (ΔQ), and the Si-Se bond of horizontal and vertical direction.

Configuration	E_ad_ (eV)	d (Å)	ΔQ (e)	Si-Se(Å)
0% NH_3_/SiSe	−0.414	2.471	0.178	2.503/2.658
−2% NH_3_/SiSe	−0.426	2.443	0.184	2.482/2.669
−2% to 0% NH_3_/SiSe	−0.413	2.454	0.180	2.505/2.661

**Table 3 sensors-20-00977-t003:** Mulliken charge transfer (ΔQ) of Si atom closest to the NH_3_ molecule and connected Se atom, the nearest distance (d) and the length of Si-Se bond of NH_3_ gas molecule adsorbed on SiSe monolayer after biaxial stress from 0% to −8%.

Parameter		0%	−2%	−4%	−6%	−8%
d (Å)		2.471	2.384	2.351	2.273	2.226
ΔQ (e)	Si	0.605	0.600	0.607	0.644	0.672
	Se	-0.618	-0.602	-0.598	-0.578	-0.568
Si-Se Bond(Å)	Horizontal	2.502	2.494	2.472	2.460	2.444
	Vertical	2.658	2.662	2.672	2.679	2.701

**Table 4 sensors-20-00977-t004:** Mulliken charge transfer (ΔQ) of Si atom closest to NH_3_ molecule and connected Se atom, the nearest distance (d) and the length of Si-Se bond of NH_3_ gas molecule adsorbed on SiSe monolayer after biaxial stress from 0% to 8%.

Parameter		0%	-2%	-4%	-6%	-8%
d (Å)		2.471	2.503	2.518	2.479	2.465
ΔQ (e)	Si	0.605	0.606	0.607	0.609	0.610
	Se	-0.618	-0.619	-0.626	-0.624	-0.627
Si-Se Bond(Å)	Horizontal	2.502	2.527	2.551	2.572	2.596
	Vertical	2.658	2.649	2.649	2.663	2.679

**Table 5 sensors-20-00977-t005:** Adsorption energy (E_ad_), Mulliken charge transfer (ΔQ), and closest distance (d) of the NH_3_ molecule and the SO_2_ molecule, respectively in SO_2_ and NH_3_/SiSe, NH_3_-SO_2_/SiSe, and SO_2_-NH_3_/SiSe configurations.

Configuration	*E_ad_* (eV)	*ΔQ*_(NH3)_(e)	*ΔQ*_(SO2)_(e)	*d*_(NH3)_(Å)	*d*_(SO2)_(Å)
SO_2_ and NH_3_/SiSe	−1.161	0.241↑	-0.276↑	2.278↓	2.646↓
NH_3_-SO_2_/SiSe	−0.752	0.243↑	-0.278↑	2.226↓	2.639↓
SO_2_-NH_3_/SiSe	−0.465	0.196↑	-0.220↓	2.431↓	2.760↑

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
