# Peer review of "Highly Selective Adsorption on SiSe Monolayer and Effect of Strain Engineering: A DFT Study"

_sensors, 2020, doi:10.3390/s20040977_

Round 1

Reviewer 1 Report

This paper describes the adsorption of several substrates to a SiSe monolayer. There are interesting results presented, but more details, particularly about the electronic structure of the material in the context of substrate binding is needed before publication. Major revisions.

English style and usage is awkward throughout. Extensive revisions are need to improve the language. The article title also needs to be improved, English-wise.

The Introduction section is too long and sounds too editorial for a scientific article.

The title lists only NH3, but a number of other substrates are examined. It seems that the title and abstract should be revised to encompass all of the results, rather than just the NH3 ones.

The CO does not interact with the monolayer, but what was the initial positioning of the molecule? If it was positioned O end down only, maybe there is a possibility that placing it C end down would have an interaction?  Did the authors try this? Similar questions could be asked about the other substrates.

Is the SiSe monolayer metallic or insulating? How do these properties change under stress? Would this have an effect on adsorption?

The discussion of the electronic structure of the material and how it changes under stress (and the orbital interactions for NH3 binding) are given short shrift. A general discussion of the band structure of the material in general should appear earlier in the paper and the binding of different substrates discussed in this context. These data seem much more interesting than the discussion of changes in density to understanding and applying the material.

Reviewer 2 Report

In the present manuscript authors report on the adsorption of small molecules on a SiSe layer, focussing the discussion on the adsorption behavior of NH3 and SO2 as known pollutants and proposing SiSe as potential candidate for the development of gas sensing devices.

Overall, the manuscript is well-organized and results are well-presented, but I have a some concerns:

-The text needs extended editing though because there are several english flaws.

- The surface model in only partially described, it is customary to give the surface cell and the lattice plane.

- Among several small molecules investigated, authors included CO2 and oxygen, major components of air mixture, but not N2. I think having the adsorption energy and site of N2 on the SiSe model layer would be useful too.

Unfortunately, the discussion on NH3/SO2 coadsorption is incomprehensible to me, because of two reasons: first, I don't understand what would be the final structure in the three cases, namely SO2&NH3/SiSe, SO2-NH3/SiSe, NH3-SO2/SiSe. Second, table 5 is unfortunately missing from the manuscript.

Reviewer 3 Report

In this paper, authors present a model for SiSe, using it to estimate the absortion of different molecules on such surface, discussing the usability of this material as a gas sensor for ammonia. Nevertheless, they do not show any physical evidence of this material being suitable for such purpose. In fact, I have not found any paper citing this material as a candidate to be used in a gas sensor application. In my opinion, it is necessary that any physical evidence must be provided to certify that the presented model really fits this material behaviour.

Moreover, English must be improved and referenes must be revised, since some of them are repeated.

Round 2

Reviewer 2 Report

The revised version of the manuscript addresses all of my former concerns.

Reviewer 3 Report

Which is the meaning of the sentence: "Yang et al. demonstrated that SiSe monolayers made from non-toxic and earth-abundant elements intrinsically"?There are other sentences in the text that must be revised.

Authors analyze the effect of stress in the adsortion capability of the SnSe layer to different molecules. How this stress can be acheieved in a real situation? Bending the substrate? Including dopant atoms in the lattice? This point should be clarified in the text, since this effect is relevant for the usability as a gas sensor

According authors analysis, NO, NO2 or SO2 once adsorbed, will remain in the surface making the system difficult to reuse. Then, if a sensor of this material is prepared, the possibility to be poisoned if exposed to thees gases is high. Therefore, this possible sensor shows an important drawback.

Taking into account the two previous points, is this materal really a good candidate for a NH3 sensor?
